# Mapping evidence of food safety at transport stations in Africa: a scoping review protocol

Busisiwe Purity Ncama,[1] Desmond Kuupiel  ,[1,2] Sinegugu E Duma,[1] Gugu Mchunu,[1] Phindile Guga,[1] Rob Slotow[3]

[1]School of Nursing and Public Health, College of Health Sciences, University of KwaZulu-Natal, Durban, South Africa
[2]Research for Sustainable Development Consult, Sunyani, Ghana
[3]School of Life Sciences, College of Agriculture, Engineering and Science, University of Kwazulu-Natal, Pietermaritzburg, South Africa

**Correspondence to**
Dr Desmond Kuupiel;
desmondkuupiel98@hotmail.com

## ABSTRACT

**Introduction** In Africa, travels, urbanisation and changing consumer habits are increasing the number of people buying and eating food prepared/sold at public spaces including transport stations, particularly in the urban and periurban areas. Although food trading in such public spaces serves as a source of livelihood for many people, unsafe food can have a negative impact on health. We, therefore, aim to systematically explore and examine the literature, and describe the evidence on food safety (food handling, storage, preparation and sale, packaging of food when sold, hygiene of sale venue and quality (nutrition) of food sold/purchased/eaten) at transport stations to inform policy, as well as identify research gaps for future studies in Africa.

**Methods and analysis** We will employ the Arksey & O'Malley framework, Levac *et al* recommendations and the Joanna Briggs Institute guidelines to guide this study. We will conduct a comprehensive search in PubMed, SCOPUS, Web of Science, Google Scholar and EBSCOhost (Academic search complete, CINAHL with Full-text and Health Source) from inception to December 2019 for relevant peer-review articles using a combination of keywords/search terms with no limitations. We will also search for relevant literature from the reference list of all included articles. Two investigators will independently screen the articles in parallel at the abstract and full-text phases using the eligibility criteria as a guide. Data extraction will be done using a piloted data extraction form designed in a Microsoft Word tabular format. Afterward, the extracted data will be collated into themes and subthemes, summarised, and the results reported using a narrative approach. We will the Preferred Reporting Items for Systematic Reviews and Meta-analyses: Extension for scoping reviews checklist to report this study results.

**Ethics and dissemination** Ethics approval is not required. All sources of data will be adequately cited and added to the reference list. We will present the final scoping review results at the appropriate workshops, meetings, conferences, as well as submit for peer-review and publication in a scientific journal.

## Strengths and limitations of this study

► This will be the first scoping review to systematically explore literature and describe the evidence on food safety at transport stations to inform policy, as well as identify research gaps for future studies in Africa.
► A comprehensive search for potentially eligible articles will be conducted in four electronic databases, and websites of international organisations.
► The scoping review permits the inclusion of all types of study designs, and it encompasses both published and grey literature.
► Boolean words AND/OR as well as medical subject heading terms will be included in the electronic search strategy to capture all relevant studies.
► This study limitation is the use of only four databases, which may potentially result in missing relevant articles indexed in other electronic databases.

substances can cause more than 200 different diseases—ranging from diarrhoea to cancers.[1] Although unsafe food can have a significant impact on health in both high-income and low-and middle-income countries (LMICs), the impact is much higher in LMICs.[2,3] Globally, it is estimated that unsafe food results in about 600 million cases of foodborne diseases and close to 420 000 deaths with about 30% of all foodborne deaths occurring among children less than 5 years annually.[1,2] The WHO further estimates that eating unsafe food results in approximately 33 million years of healthy lives lost worldwide every year.[1] As a result, in 2006, WHO declared foodborne diseases as well as food-related diseases and food safety as a major public health concern.[4,5] Unsafe food creates a vicious cycle of disease and malnutrition, particularly affecting infants, young children, the elderly and the sick.[6] To this end, access to safe and nutritious and sufficient food is key to sustaining life and promoting good health.

Food safety may denote the prevention of biological or non-biological hazards in food, whether chronic or acute that may make food

## BACKGROUND

Food is one of the major modes of disease transmission for microbial, chemical and physical hazards. Food containing harmful bacteria, viruses, parasites or chemical

harmful to the health of the consumer.[7] Food safety is about producing, handling, storing and preparing food in such a way as to prevent infection and contamination in the food production chain, and to help ensure that food quality and wholesomeness are maintained to promote good health.[7] In addition to contributing to food and nutrition security, a safe food supply also supports national economies, trade, and tourism, stimulating sustainable development.[7 8] The globalisation of food trade, a growing world population, climate change and rapidly changing food systems have an impact on the safety of food.[7 8]

In most LMICs, food trading at transport stations such as taxi ranks, bus stations and lorry parks is fast-growing, particularly in the urban and periurban areas. It is evident that over 2.4 billion people eat food sold by vendors at public spaces including transport stations every day worldwide.[9 10] This may be due to several reasons such as the thinking that the foods sold in public spaces are appealing, ready-to-eat, convenience and of low cost and savoury taste.[11–13] Food trading in public spaces serves as a source of livelihood for many households in LMICs.[8 11 13] Although the food sold at transport stations may also have great nutritional value depending on the type of ingredients and the conditions under which the food was prepared and served to the customer, it may as well cause ill health arising from contamination of the food. Since the food is most often prepared and sold in an environment that often lacks sanitary facilities and health surveillance, and at times, not regulated by the appropriate local authorities in most LMICs.[13–15] Despite this, no study to the best of our knowledge has mapped literature on food safety at transport stations in Africa. We, therefore, intend to systematically search for literature, examine the scope of the literature and describe the evidence on food safety at transport stations in Africa in order to inform policy. We also anticipate identifying gaps from the study findings for future research to improve food safety at transport stations such as taxi ranks, bus stations and lorry parks in Africa.

## METHODS
### Design of the study protocol
We will employ an amended Arksey & O'Malley framework incorporating the Levac et al recommendation[16 17] to guide this scoping review. The Arksey & O'Malley framework includes the following: research question identification; identifying relevant studies; selection of study; data charting, collating, summarising and reporting the findings, and consultation. However, we will exclude the consultation stage because this review is nested in a larger study that already includes stakeholder consultation. We followed the Preferred Reporting Items for Systematic Reviews and Meta-Analyses Extension for Protocols (PRISMA-P) guidelines to develop this protocol (online supplementary file 1). However, we will follow the PRISMA:

**Table 1** PCC framework for defining the eligibility of the studies for the primary research question

| P- Population | Transport stations: This will include bus stations, lorry parks, taxi ranks and automobile stations and bus stops. |
|---|---|
| C-Concept | Food safety: This includes food handling, storage, preparation and sale, packaging of food when sold, hygiene of sale venue and quality (nutrition) of food sold/purchased/eaten. |
| C-Context | African |

Extension for scoping reviews checklist to report this study results.

### Identifying the research question
This scoping review question shall be:

What is the scope of evidence on food safety at transport stations in Africa? Table 1 presents the Population, Concept and Context mnemonic used to inform the eligibility of this proposed review.[18]

### Identify relevant studies
We will conduct a comprehensive search in PubMed, SCOPUS, Google scholar, Web of Science and EBSCO-host (Academic search complete, CINAHL with Full-text and Health Source) from inception to December 2019 for relevant articles. We will develop the search strategy in consultation with a subject Librarian at the University of KwaZulu-Natal so as to enable us capture all/most relevant articles for the study. We will use a combination of the following keywords: "transport station" OR "taxi rank" OR "automobile station" OR "bus station" OR "lorry park" OR "car park" OR "transport hub" OR "streets" OR "public space" AND "food safety" OR "food handling" OR "food storage" OR "food preparation" OR "food packaging" OR "food sale" OR "food vend" OR "food sale" OR "food sold" OR "food hawking" OR "food purchase" in order to enable us capture all appropriate peer-review articles. We will include Booleans terms AND/OR and medical subject heading terms in our search. Date, study design and language limitations will be removed during the databases search. We will also search the reference list of all included studies for articles presenting useful information to address this proposed scoping review question. The search date, database, keywords, number of retrieved articles, number of eligible articles and all other relevant search records will be documented appropriately in detail. Table 2 illustrates a full-search strategy in PubMed database for this proposed scoping review study.

### Eligibility criteria and study selection
The eligibility criteria will be as outlined below.

#### Inclusion criteria
This shall include the following:
- ► Studies reporting evidence from Africa.
- ► Studies presenting evidence on food safety.

**Table 2** A full-search strategy in PubMed illustrating the feasibility of the proposed scoping review

| Date | Database | Keywords | Search results |
|---|---|---|---|
| 6 April 2020 | PubMed | (("transport"[All Fields]) AND station[All Fields]) OR (taxi[All Fields] AND rank[All Fields]) OR (("automobiles"[MeSH Terms] OR "automobiles"[All Fields] OR "automobile"[All Fields]) AND station[All Fields]) OR "bus station"[All Fields] OR (("motor vehicles"[MeSH Terms] OR ("motor"[All Fields] AND "vehicles"[All Fields]) OR "motor vehicles"[All Fields] OR "lorry"[All Fields]) AND park[All Fields]) OR "car park"[All Fields] OR "transport hub"[All Fields] OR "streets"[All Fields] OR "public space"[All Fields] AND "food safety"[All Fields] OR "food handling"[All Fields] OR "food storage"[All Fields] OR "food preparation"[All Fields] OR "food packaging"[All Fields] OR "food sale"[All Fields] OR (("food"[MeSH Terms] OR "food"[All Fields]) AND vend[All Fields]) OR "food sale"[All Fields] OR "food sold"[All Fields] OR (("food"[MeSH Terms] OR "food"[All Fields]) AND ("hawks"[MeSH Terms] OR "hawks"[All Fields] OR "hawking"[All Fields])) OR "food purchase"[All Fields] AND (("0000/01/01"[PDAT] : "2019/12/31"[PDAT]) AND "humans"[MeSH Terms]) | 13644 |

► Studies reporting from transport stations.
► Studies published in English.
► Primary studies focusing on food safety at transportations.

### Exclusion criteria
We shall exclude the following:
► Studies presenting evidence from other countries outside Africa.
► Studies presenting evidence of food safety from other sites such as homes, restaurants, markets and hotels, and schools.
► Studies published in other languages such as French, Arabic and Portuguese.
► Review studies.

### Study selection
The screening for potentially eligible studies will be done in three phases. In phase 1, a comprehensive article search in the electronic databases will be conducted by one investigator and import all eligible articles to a new endnotes X9 library created for the review. To clean the library, all duplicates will be deleted and subsequently shared with the coinvestigators. In phase 2, two investigators will independently screen the abstracts in parallel guided by the eligibility criteria designed using a google form. All disagreements between the investigators in terms of their response at this stage will be addressed through discussions among the investigators until a consensus is reached. Similarly, in the third phase, two investigators will independently screen all the full-text articles as described in phase 2, however, a third investigator will be engaged to resolve any discrepancies in their responses. Cohen's kappa statistic will be calculated following full-text screening to demonstrate the level of agreement between reviewers. All articles retrieved from the reference list of included studies will also be screened using the same process. Where a full-text article cannot be retrieved or not accessible from the online databases, we will seek assistance from the University of KwaZulu-Natal library or write to the authors to request the full

text for screening. An amended PRISMA flow diagram will be used to report the search and screening results[19] (figure 1).

### Charting the data
We will design a data extraction form using Microsoft Word in a tabular format. To ensure the trustworthiness and reliability of the data, two independent investigators will pilot the data extraction form using a random sample of 10% of the included articles in parallel. Subsequently, the data extraction form will be revised if necessary, to ensure its rigour and ability to capture all relevant data to answer the review question. We will keep the data extraction form updated throughout this process until all relevant information has been extracted from the included articles. Box 1 shows the data extraction form that will be used for this scoping review study.

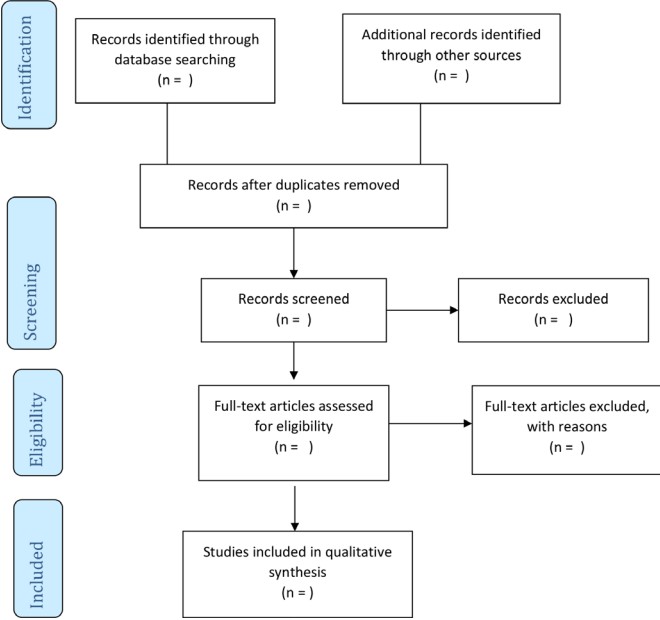

**Figure 1** PRISMA 2009 flow diagram.[19] PRISMA, Preferred Reporting Items for Systematic Reviews and Meta-Analyses.

## Box 1  Data extraction form

Author and publication year.
Study title.
Study aim/objective.
Type of study design.
Study setting (country).
Study participants.
Significant findings.
Food safety outcomes.
Conclusions/recommendations.

## Collating, summarising and reporting the results

We will present information about the included articles that are aligned with the research question of this proposed scoping review study. All relevant data from the eligible studies will be extracted using the piloted data extraction form to answer the review question. Both deductive (for characteristics of the included articles) and inductive (for study findings/outcomes) approaches will be used to extract relevant data. Thematic analysis will be utilised to collate the study findings into themes and sub-themes. Then, a narrative summary of the themes and subthemes will be reported. To ensure rigour, the analysis will be conducted by two investigators with input from collaborators. The PRISMA extension designed for the scoping reviews checklist will be used to guide the reporting of the results of this scoping review. Afterward, a meta-analysis may be conducted using quantitative data if possible.

## Ethics and dissemination

Not required. All sources of data will be adequately referenced. We will present the final scoping review results at the appropriate workshops, meetings, conferences, as well as submit for peer-review and publication in a scientific journal.

**Contributors**  BPN, DK, SED, GM and RS conceptualised the study. DK wrote the manuscript and PG contributed to the writing. BPN, SED, GM and RS critically review the manuscript and made revisions. DK wrote the final draft manuscript and all the authors approved it.

**Funding**  Funding for this work was provided by the Sustainable and Healthy Food Systems (SHEFS) Programme, supported through the Wellcome Trust's Our Planet, Our Health Programme (grant number: 205200/Z/16/Z).

**Disclaimer**  The funder played no role in the literature search and writing of the manuscript.

**Competing interests**  None declared.

**Patient and public involvement**  Patients and/or the public were not involved in the design, or conduct, or reporting, or dissemination plans of this research.

**Patient consent for publication**  Not required.

**Provenance and peer review**  Not commissioned; externally peer reviewed.

**ORCID iD**
Desmond Kuupiel http://orcid.org/0000-0001-7780-1955

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
