## [Reviewer comments · BMJ Open]

ARTICLE DETAILS

TITLE (PROVISIONAL)	Mapping Evidence of Food Safety at Transport stations in Africa: A Scoping Review Protocol
AUTHORS	Ncama, Busisiwe; Kuupiel, Desmond; Duma, Sinegugu; Mchunu, Gugu; Guga, Phindile; Slotow, Rob

VERSION 1 – REVIEW

REVIEWER	Warren Dodd University of Waterloo, Canada
REVIEW RETURNED	28-Jan-2020

GENERAL COMMENTS	This protocol provides an overview of a planned scoping review that will systematically map the existing literature on food safety at transport stations in Africa. Particularly strong within this scoping review protocol is the rationale for why such a review is needed. Some comments and questions for the authors to consider are below: Overall 1. It would be helpful if dates were included for the study. When will the review begin? Are there date restrictions associated with the sources included in the review? Search strategy 2. Some rationale could be provided to justify the use of Google Scholar over another database.3. I am flagging for the authors (although it appears the authors are aware) that the search string will need to be modified for each database used4. Is there a particular strategy that will be applied in using Google to scan the relevant grey literature? For example, the authors could integrate lessons from: Godin, K., Stapleton, J., Kirkpatrick, S. I., Hanning, R. M., & Leatherdale, S. T. (2015). Applying systematic review search methods to the grey literature: a case study examining guidelines for school-based breakfast programs in Canada. Systematic reviews, 4(1), 138.5. Some more detail could be provided in terms of how the search string will need to be modified to search the relevant grey literature including the WHO and FAO websites. Is there a particular section of the website(s) that will be accessed for relevant content? Inclusion/exclusion criteria
--

	6. As part of the inclusion/exclusion criteria, will language of the article be a factor in determining eligibility of studies/resources? (e.g., only including English publication?) 7. The issue of 'health-related' issues is vague, and I can anticipate some confusion associated with discerning when an article discusses a 'health-related' issue. For example, food security is sometimes discussed in terms of health and sometimes discussed in terms of broader socio-cultural contexts. Some more clarity around these inclusion/exclusion criteria might help the authors in the discernment process. Study selection 8. With Cohen's kappa be calculated at each stage of the screening process to demonstrate the level of agreement between investigators? Collating, summarizing, and reporting the results 9. More detail and justification could be provided around the proposed 'narrative approach' to present results. Why is this approach better than thematic analysis or some other way of collating themes? Will this process be completed by one investigator or collaboratively completed by several investigators? Will this narrative approach be informed by a deductive or inductive approach?
--	--

REVIEWER	Sizwe Makhunga University of KwaZulu-Natal, South Africa
REVIEW RETURNED	18-Feb-2020

GENERAL COMMENTS	 1. According to the title, the study will focus on food safety at transport stations. However, ln110 p4; ln125 p5 and ln139 p5 suggest that it will be public spaces. Provide a brief description of "public spaces". 2. Please check grammatical errors. 3. ln87 p4 - did you mean for? 4. ln88 p4 food containing harmful bacteria etc. is unsafe. You can remove "unsafe" 5. ln89 p4 Please check if the statement is true always. In most cases the viral load or microbial load is the cause. 6. Please double check the statement in ln101/102 p4. 7. ln110 p4 Please decide on whether the focus of the study is public spaces or transport stations. 8. ln118 p4 what health implications? 9. ln119 p5 surveillance by who? By legal control, do you mean not regulated? If so by who? 10. ln125 food safety in public places or transport stations? 11. ln139 p5 Did you mean food safety in public spaces. 12. ln140 p5 the population in Table 1 is different from that in ln139 p5. 13. ln166 p6 state the study population - public spaces or transport stations 14. ln168 p6 food safety and/or security 15. ln168 very vague. 16. ln169 p6 primary studies on what? You cannot include all primary studies about everything. 17. Revisit ln175 p6. The statement is broad. What is meant about 'others' 18. ln176 p6 Why?
---

	19. In169 p6 vague statement. What are non-health topics? Is the study not on food safety? 20. In183 p6 Did you mean 'eligible' studies? Why new endnote? 21. In191 p7 collaborate..to do what? 22. In122 p7 Published...either English? 23. In212 p7 summarise? Is the study going to use a thematic or narrative approach? 24. In214 p7 Remove the statement about meta-analysis because this is solely a scoping review. 25. in218 ...to inform for food safety?
--	--

VERSION 1 – AUTHOR RESPONSE

Responses to Reviewer #1 (Warren Dodd) comments

General comment

Please leave your comments for the authors below

This protocol provides an overview of a planned scoping review that will systematically map the existing literature on food safety at transport stations in Africa. Particularly strong within this scoping review protocol is the rationale for why such a review is needed. Some comments and questions for the authors to consider are below:

Response

We are most grateful to you for finding time out of your busy schedule to review this manuscript. Kindly find below a detailed response to the comments for your consideration.

Comment 1

It would be helpful if dates were included for the study. When will the review begin? Are there date restrictions associated with the sources included in the review?

Response

Although the review is at the stage of preliminary searches, we anticipate starting the review as soon as this protocol has been accepted. This will allow us to incorporate views or suggestions from the reviewers for a thorough study. In view of this, we are unable to provide a definite starting date. However, this study will be limited to studies published from inception to December 2019 as indicated in the abstract (Page 2, LN 43-44) and Page 5, LN 151.

Search strategy

Comment 2

Some rationale could be provided to justify the use of Google Scholar over another database.

Response

Thank you for your comment. We have revised this section. Please see page 2, LN 42-43 and page 5, LN 149-150.

Comment 3

I am flagging for the authors (although it appears the authors are aware) that the search string will need to be modified for each database used

Response

We will develop the search strategy in consultation with a subject librarian at the University of KwaZulu-Natal so as to enable us capture all/most relevant articles for the study (Page 5, LN 151-153).

Comment 4

Is there a particular strategy that will be applied in using Google to scan the relevant grey literature? For example, the authors could integrate lessons from:

Godin, K., Stapleton, J., Kirkpatrick, S. I., Hanning, R. M., & Leatherdale, S. T. (2015). Applying systematic review search methods to the grey literature: a case study examining guidelines for school-based breakfast programs in Canada. *Systematic reviews*, 4(1), 138.

Response

We are most grateful for your suggestion however, we have revised this section as indicated in our response to your second comment

Comment 5

Some more detail could be provided in terms of how the search string will need to be modified to search the relevant grey literature including the WHO and FAO websites. Is there a particular section of the website(s) that will be accessed for relevant content?

Response

Thank you. Please we have revised this as indicated in our response to comment 6.

Inclusion/exclusion criteria

Comment 6

As part of the inclusion/exclusion criteria, will language of the article be a factor in determining eligibility of studies/resources? (e.g., only including English publication?)

Response

Yes, please. Kindly page 6, LN 168-182 for the revised inclusion and exclusion criteria.

Comment 7

The issue of 'health-related' issues is vague, and I can anticipate some confusion associated with discerning when an article discusses a 'health-related' issue. For example, food security is sometimes discussed in terms of health and sometimes discussed in terms of broader socio-cultural contexts. Some more clarity around these inclusion/exclusion criteria might help the authors in the discernment process.

Response

We sincerely agree with you. Kindly page 6, LN 168-182 for the revised inclusion and exclusion criteria.

Study selection

Comment 8

With Cohen's kappa be calculated at each stage of the screening process to demonstrate the level of agreement between investigators?

Response

Yes, please. Kindly see page 7 LN 194-196 for confirmation.

Collating, summarizing, and reporting the results

Comment 9

More detail and justification could be provided around the proposed 'narrative approach' to present results. Why is this approach better than thematic analysis or some other way of collating themes? Will this process be completed by one investigator or collaboratively completed by several investigators? Will this narrative approach be informed by a deductive or inductive approach?

Response

We have revised this section. Please pages 7 and 8, LN 215-220 for your consideration.

Responses to Reviewer #2 (Sizwe Makhunga) comments

General response

We are most grateful to you for finding time out of your busy schedule to review this manuscript. Kindly find below a detailed response to the comments for your consideration.

Comment 1

According to the title, the study will focus on food safety at transport stations. However, In110 p4; In125 p5 and In139 p5 suggest that it will be public spaces. Provide a brief description of 'public spaces'.

Response

We have generally addressed this concern in the revised manuscript. Kindly see examples page 4, LN 112, and page 5, LN 126 and 128 for your confirmation.

Comment 2

Please check grammatical errors.

Response

We have proofread this manuscript again and all grammatical errors identified have been corrected.

Comment 3

In87 p4 - did you mean for?

Response

Yes, please. We have addressed it (Page 4, LN 89).

Comment 4

In88 p4 food containing harmful bacteria etc. is unsafe. You can remove "unsafe"

Response

We have revised it. Please page 4, LN 90.

Comment 5

In89 p4 Please check if the statement is true always. In most cases the viral load or microbial load is the cause.

Response

Thank you. You may be right depending on the virulence of the microbial agent. However, the mere presence of microbes in food can generally cause some illnesses although the severity may not be the same. So we are convinced the statement referred to is also true. Please refer to the link below for confirmation.

<https://www.who.int/activities/estimating-the-burden-of-foodborne-diseases>.

Comment 6

Please double check the statement in In101/102 p4.

Response

Please we have double checked it and statement revised (Page 4, LN 103-104).

Comment 7

In110 p4 Please decide on whether the focus of the study is public spaces or transport stations.

Response

We have addressed this concern. Please page 4, LN112.

Comment 8

In118 p4 what health implications?

Response

We have revised the statement (Pages 4 and 5, LN 120-121).

Comment 9

In119 p5 surveillance by who? By legal control, do you mean not regulated? If so by who?

Response

We have revised the statement (Page 5, LN 122-123).

Comment 10

In125 food safety in public places or transport stations?

Response

Please we have addressed it (Page 5, LN 126).

Comment 11

In139 p5 Did you mean food safety in public spaces.

Response

We have addressed it (Page 5, LN 139).

Comment 12

In140 p5 the population in Table 1 is different from that in In139 p5.

Response

Thank you. We have addressed this please (Page 5, LN 144).

Comment 13

In166 p6 state the study population - public spaces or transport stations

Response

Please we have addressed this concern throughout the manuscript as earlier stated

Comment 14

In168 p6 food safety and/or security

Response

We have revised the inclusion criteria. Please page 6, LN 168-182.

Comment 15

In168 very vague.

Response

We have revised the inclusion criteria as indicated above (Page, LN 168-182).

Comment 16

In169 p6 primary studies on what? You cannot include all primary studies about everything.

Response

Thank you. But we have previously indicated our study population and concept hence, we found unnecessary to repeat it. Nonetheless, we will include primary studies focusing on food safety at transportations. Please see the revised inclusion criteria.

Comment 17

Revisit In175 p6. The statement is broad. What is meant about 'others'

Response

We have revised the exclusion criteria. Please page 6 (Page 6, LN 168-182).

Comment 18

In176 p6 Why?

Response

We have revised the exclusion criteria. Please page 6, LN168-182).

Comment 19

In169 p6 vague statement. What are non-health topics? Is the study not on food safety?

Response

We have revised the exclusion criteria. Please page 6, LN 168-182.

Comment 20

In183 p6 Did you mean 'eligible' studies? Why new endnote?

Response

Yes, please. The new library will be specifically for this study. This is good practice because it helps one to export retrieved articles from the databases to the right library, screen the appropriate articles and account accurately.

Comment 21

In191 p7 collaborate..to do what?

Response

We have removed this statement and the inclusion and exclusion criteria revised

Comment 22

In122 p7 Published...either English?

Response

LN 122 does not exist on page 7 but we guessed you were referring to the previous LN 192 where the word "than" was omitted. Nonetheless, we have addressed it as indicated above.

Comment 23

In212 p7 summarise? Is the study going to use a thematic or narrative approach?

Response

We have revised this section. Please pages 7 and 8 LN 215-220 for your consideration.

Comment 24

In214 p7 Remove the statement about meta-analysis because this is solely a scoping review.

Response

Thank you for your comment however, the authors are simply demonstrating how quantitative data will be handled in future since statistical analysis will not be conducted in this proposed review. Therefore, we still find the statement appropriate and relevant.

Comment 25

in218 ...to inform for food safety?

Response

Thanks for drawing our attention to this error. We have deleted the word “for”.

VERSION 2 – REVIEW

REVIEWER	Warren Dodd University of Waterloo Canada
REVIEW RETURNED	21-Apr-2020

GENERAL COMMENTS	The authors have responded to my comments and I am satisfied with their responses and the incorporation of feedback into the protocol.
--

REVIEWER	Sizwe Makhunga University of KwaZulu-Natal
REVIEW RETURNED	27-Apr-2020

GENERAL COMMENTS	The authors have doubled. Some even conceptualized the study on a second submission. Please explain this.
---